# Metabolism of the Flavonol Kaempferol in Kidney Cells Liberates the B-ring to Enter Coenzyme Q Biosynthesis

**DOI:** 10.3390/molecules25132955

**Published:** 2020-06-27

**Authors:** Lucía Fernández-del-Río, Eric Soubeyrand, Gilles J. Basset, Catherine F. Clarke

**Affiliations:** 1Department of Chemistry and Biochemistry and the Molecular Biology Institute, University of California, Los Angeles, CA 90095, USA; 2Department of Horticultural Sciences, University of Florida, Gainesville, FL 32611, USA; esoubeyrand@gmail.com (E.S.); gbasset@ufl.edu (G.J.B.)

**Keywords:** flavonoids, flavonol, kaempferol, coenzyme Q, kidney cells, precursor

## Abstract

Coenzyme Q (CoQ) is an essential component of the mitochondrial electron transport chain and an important antioxidant present in all cellular membranes. CoQ deficiencies are frequent in aging and in age-related diseases, and current treatments are limited to CoQ supplementation. Strategies that rely on CoQ supplementation suffer from poor uptake and trafficking of this very hydrophobic molecule. In a previous study, the dietary flavonol kaempferol was reported to serve as a CoQ ring precursor and to increase the CoQ content in kidney cells, but neither the part of the molecule entering CoQ biosynthesis nor the mechanism were described. In this study, kaempferol labeled specifically in the B-ring was isolated from *Arabidopsis* plants. Kidney cells treated with this compound incorporated the B-ring of kaempferol into newly synthesized CoQ, suggesting that the B-ring is metabolized via a mechanism described in plant cells. Kaempferol is a natural flavonoid present in fruits and vegetables and possesses antioxidant, anticancer, and anti-inflammatory therapeutic properties. A better understanding of the role of kaempferol as a CoQ ring precursor makes this bioactive compound a potential candidate for the design of interventions aiming to increase endogenous CoQ biosynthesis and may improve CoQ deficient phenotypes in aging and disease.

## 1. Introduction

Coenzyme Q (CoQ or ubiquinone) is a small lipophilic molecule found ubiquitously in cell membranes. Structurally, it is composed of a benzoquinone ring and a polyisoprenoid tail that varies in length between species [1]. In mammals, CoQ_9_ (nine-isoprene tail) and CoQ_10_ (ten-isoprene tail) are present, with CoQ_9_ predominant in rodents and CoQ_10_ in humans [1]. CoQ synthesis occurs within mitochondria through multiple steps carried out by, at least, 14 proteins known as COQ proteins [1,2]. CoQ plays a role in multiple cellular functions [3,4]. However, the primary function of CoQ is to accept electrons and protons from the respiratory complexes I and II and donate them to complex III [1,4]. This redox capacity allows CoQ to cycle between three different states: ubiquinone (oxidized), semiquinone (semi-oxidized), and ubiquinol (reduced) [1,5]. In its ubiquinol form, CoQH_2_ plays an important antioxidant role and provides protection to DNA, proteins, and lipids against oxidative stress [6,7].

CoQ content decreases with age in a variety of mammalian tissues, as reflected by a decreased biosynthesis rate [8,9,10]. The possibility of increasing CoQ_10_ content through dietary supplementation has been widely explored in recent decades [6,9]. Although more controlled studies are needed to determine the effectiveness of CoQ_10_ as an anti-aging drug in humans [11], it was previously reported that CoQ plasma levels in the elderly correlate with enhanced physical activity and lower lipid oxidative damage; and that CoQ_10_ supplementation improves vitality, physical performance, and quality of life in old individuals [9]. The case for beneficial effects of CoQ_10_ supplementation is stronger for a number of age-related diseases such as cardiovascular diseases, neuropathies, inflammation, metabolic syndrome, arthritis, carcinogenesis, diabetes, osteoporosis, and hypercholesterolemia [3,8,11]. CoQ_10_ supplementation has been shown to reduce inflammatory markers, which are commonly present at high levels in the aforementioned aged-related diseases [12,13,14].

However, the long polyisoprenoid chain makes CoQ_10_ highly lipophilic and difficult to absorb. CoQ_10_ dietary supplements pose several challenges, including specific absorption via the gastrointestinal tract [5], cellular uptake at the plasma membrane, transport across intracellular membranes, and assimilation by mitochondria. All these trafficking steps render the process of exogenous CoQ_10_ supplementation very inefficient [3,9]. In this regard, alternative vehicles for CoQ_10_ administration are under study (e.g., oil-based capsules, nanoparticles) [3,15,16], as well as novel strategies that could potentiate the endogenous synthesis of CoQ [2,3]. Previously, we described the ability of kaempferol, a flavonol found in fruits and vegetables, to increase CoQ content by acting as a novel CoQ precursor in mouse and human kidney cells [17]. However, the exact metabolic pathway by which kaempferol participates in CoQ biosynthesis was not identified. Two hypotheses were proposed: (1) Kaempferol could be a direct substrate of COQ2 in the CoQ biosynthetic pathway and would be subsequently metabolized by the different COQ proteins until it reached the final structure of CoQ; or alternatively, (2) kaempferol could be metabolized in the cell to produce a potential CoQ ring precursor, which would be then integrated in the CoQ biosynthetic pathway [17]. In a recent study, Soubeyrand and co-authors [18] described that, in plants, the biosynthetic pathways of flavonoids and CoQ are indeed connected and that kaempferol can serve as a precursor for the synthesis of CoQ. They proved that the B-ring of kaempferol is subjected to peroxidative cleavage, to give 4-hydroxybenzoic acid (4HB), a common precursor of the benzoquinone ring of CoQ [18].

The goal of the present work is to further describe the relationship between kaempferol and CoQ in mammalian cells. Our results show that in kidney cells, the B-ring of kaempferol is the part of the molecule that enters CoQ biosynthesis, suggesting that the mechanism described for plants is likely to be conserved in vertebrates.

## 2. Results

To further understand how kaempferol functions as a CoQ precursor in mammalian cells, we decided to test whether the B-ring of kaempferol is the part of the molecule that enters into the CoQ biosynthetic pathway, as was reported to occur in plants [18]. Our efforts to chemically synthesize kaempferol specifically ^13^C-labeled in the B-ring (^13^C_6_*-[B-ring]-*kaempferol) were unsuccessful. As an alternative strategy, we chose to isolate ^13^C_6_*-[B-ring]-*kaempferol from cultures of the plant *Arabidopsis thaliana*. Such an in vivo synthesis of B-ring labeled kaempferol is possible because plants derive the B-ring of kaempferol exclusively from the phenyl moiety of phenylalanine [19]. In contrast, the A-ring and C-ring originate from malonyl-CoA [19]. By feeding ^13^C_6_*-*L-phenylalanine (^13^C_6_-Phe) to *Arabidopsis* plants grown in sterile conditions, one can therefore obtain kaempferol specifically labeled on the B-ring [18]. Furthermore, to boost kaempferol accumulation, the ^13^C_6_-Phe feeding was performed using a *flavonoid-3′-hydroxylase Arabidopsis* knockout, which cannot further metabolize kaempferol into anthocyanins [20]. One should note that kaempferol obtained with such a method consists in a mixture of ^13^C_6_*-[B-ring]-*kaempferol as well as unlabeled kaempferol, which was present in the plant tissues prior to the feeding with ^13^C_6_-Phe. The specific enrichment of ^13^C_6_*-[B-ring]-*kaempferol in the mixture used for our experiments was approximately 10% of the total pool of kaempferol (i.e., unlabeled+labeled).

Using ^13^C_6_*-[B-ring]-*kaempferol extracted and purified from *Arabidopsis*, we treated mouse kidney proximal tubule epithelial (TKPTS) cells and measured de novo and total content of CoQ (Figure 1). Unlabeled kaempferol, universally ^13^C-labeled kaempferol (^13^C-kaempferol) and ^13^C-labeled 4HB (^13^C_6_-4HB) were used as complementary treatments (Figure 1a). Cells treated with ethanol vehicle were included as a control. We observed that in terms of total CoQ (CoQ+^13^C_6_-CoQ), both CoQ_9_ and CoQ_10_ content were increased by treatment with kaempferol (independently of the label) and 4HB (Figure 1b,c), as described previously for kidney cells [17]. ^13^C_6_-CoQ was detected in cells treated with ^13^C-kaempferol and ^13^C_6_-4HB, which is in agreement with the role of these compounds as CoQ ring precursors [17,21]. Notably, treatment with ^13^C_6_*-[B-ring]-*kaempferol also led to synthesis of ^13^C_6_-CoQ (Figure 1b,c), indicating that the B-ring of kaempferol is the part of the molecule that enters CoQ biosynthesis. As expected, the lower specific labeling of the B-ring in the ^13^C_6_*-[B-ring]-*kaempferol/kaempferol mixture resulted in the lower amount of ^13^C_6_-CoQ (Figure 1b,c).

## 3. Discussion

Kaempferol is a natural flavonol-type flavonoid present in tea as well as in numerous vegetables and fruits such as broccoli, grapes, kale, tomatoes, and citrus fruits, among others [22,23]. The most well-known properties of kaempferol are its anti-inflammatory effects in both acute and chronic inflammation, and its role in the prevention of multiple types of cancer [24,25,26]. Moreover, it has been demonstrated to protect liver and heart function and to prevent metabolic and neurodegenerative diseases [24,26]. Kaempferol is thought to achieve its beneficial effects through the regulation of a multitude of cellular pathways [24,26], but its antioxidant function might also be important. Kaempferol significantly lowers oxidative stress and lipid peroxidation and can improve antioxidant defense activity [26]. The C-3 hydroxyl group has been considered especially important for this antioxidant activity [27].

In 2015, Xie et al. [21] described that the dietary compound resveratrol, which has been linked to multiple health benefits [28], can serve as ring precursor in the biosynthesis of CoQ in *Escherichia coli*, *Saccharomyces cerevisiae,* and mammalian cells. In a later study, kaempferol was additionally described to act as a CoQ ring precursor, and to increase CoQ content in mammalian cells [17]. The increase of CoQ induced by kaempferol was proven to be stronger than the effect exerted by other polyphenols, including resveratrol. In fact, quercetin, naringenin, luteolin, and piceatannol did not show any effect [17]. These studies linked natural products generally present in the diet with CoQ biosynthesis, although the mechanism responsible for the incorporation was not determined. Recently, it was described that the synthesis of flavonoids is indeed connected to the synthesis of CoQ in plants [18]. Moreover, the authors showed that in plants, the B-ring of kaempferol is cleaved by still-unknown peroxidases producing 4HB that directly enters into the CoQ biosynthesis pathway [18].

Here, we confirmed that similar enzymes might be present in mammalian, cells allowing for a comparable mechanism to occur. Although additional studies are necessary to determine whether the cleavage of kaempferol in mammals produces 4HB, our results prove that the B-ring of the flavonol is the part of the molecule that enters into the CoQ biosynthetic pathway (Figure 2). These specific enzymes must be shared in plants and mammalian cells but seem to be absent in *S. cerevisiae*, at least in the BY4741 genetic background, since yeast were described to incorporate ^13^C-kaempferol very marginally [17]. In plants, the chemistry of the reaction requires the simultaneous presence of a double bond between C-2 and C-3, and a hydroxyl group on C-3 [18], since dihydrokaempferol (no C2-C3 double bond) and naringenin (no C2-C3 double bond nor C-3 –OH) failed to be substrates of the peroxidative cleavage. The previous independent observation that apigenin (no C2-C3 double bond) and naringenin failed to enhance CoQ content in kidney cells [17] supports the hypothesis that plants and mammals share a similar mechanism.

Given the limited bioavailability of CoQ_10_ supplements, the stimulation of the endogenous synthesis of CoQ has been the focus of several studies [1,9]. Understanding how kaempferol augments the CoQ biosynthetic pathway is of outstanding importance, since its capacity to increase endogenous CoQ content has a strong potential to ameliorate CoQ deficiencies associated with aging or disease. Moreover, patients could find additional benefits since the regular consumption of flavonoids is related to a reduced risk of age-related diseases as described above [24,25,26]. Although the bioavailability of kaempferol is quite low [29], the increase in CoQ in kidney cells was observed at doses that can be attainable physiologically, by oral supplementation or by consumption of flavonoids-containing food [27], and even a slight supplementary amount of CoQ precursors could move the metabolic flux in favor of CoQ synthesis.

Additional research is needed to characterize kaempferol as an efficient compound for the treatment of CoQ deficiencies. Further in vitro and in vivo studies are necessary to fully understand the relationship between kaempferol and CoQ, find the most suitable formulation of the bioactive compound, and identify the enzyme(s) responsible for the peroxidative cleavage.

## 4. Materials and Methods

### 4.1. Chemicals and Reagents

Non-labeled kaempferol was obtained from Santa Cruz Biotechnology, Inc. (Dallas, TX, USA); ^13^C_6_-4HB from Cambridge Isotope Laboratories, Inc. (Tewksbury, MA, USA); and ^13^C-kaempferol from Isolife (Wageningen, The Netherlands). CoQ_9_ and CoQ_10_ standards were obtained from Sigma-Aldrich (San Luis, MO, USA). Dipropoxy-CoQ_10_ was synthesized essentially as described by Edlund [30] for diethoxy-Q_10_, except 1-propanol was substituted for ethanol while maintaining the other reagents and conditions. ^13^C_6_*-[B-ring]-*kaempferol was prepared from in vitro cultures of *Arabidopsis thaliana flavonoid-3′-hydroxylase* knockout plants fed for 48 h with 250 µM doses of ^13^C_6_*-*L-Phenylalanine (Cambridge Isotope Laboratories, Inc., Tewksbury, MA, USA) [18]. Leaves (~1.5 g) were homogenized using a Pyrex tissue grinder in 5 × 900 μL of methanol, and the extracts were centrifuged at 18,000× *g* for 10 min. The supernatants (5 × ~800 μL) were pooled and mixed to an equal volume of 2 M HCl and incubated at 70 °C for 40 min in order to hydrolyze the glycosyl–kaempferol conjugates. Hydrolysate aliquots (200 μL) were mixed with an equal volume of 100% methanol and centrifuged at 18,000× *g* for 15 min. Samples (100 μL each) were chromatographed on a Zorbax Eclipse Plus C18 column (4.6 × 100 mm, 3.5 µm; Agilent Technologies, Santa Clara, CA, USA) held at 30 °C using a 25-min linear gradient starting from 10 mM ammonium formate pH 3.5 to 100% methanol at a flow-rate of 0.8 mL/min. Kaempferol (18.7 min) was collected by monitoring the absorbance at 365 nm, evaporated to dryness with nitrogen gas, and then resuspended in 100% methanol for quantification using a molar extinction coefficient of 21,242 M^−1^ cm^−1^. MS/MS analyses indicated that the preparation was composed of ~10% of ^13^C labeled kaempferol (M + 6) and ~90% of unlabeled kaempferol.

### 4.2. Cell Culture Conditions and Treatments

Mouse kidney proximal tubule epithelial (TKPTS) cells [31], were provided by Dr. Elsa Bello-Reuss (Texas Tech University Health Science Center, Lubbock, TX, USA) and Dr. Judit K. Magyesi (University of Arkansas for Medical Sciences, Little Rock, AR, USA). TKPTS cells were grown in DMEM/F12 containing 4.5 g/L glucose, and supplemented with 10% fetal bovine serum (FBS), 2 mM L-glutamine, and gentamicin–amphotericin B (125 µg/mL and 5 mg/mL, respectively). Cultures were maintained at 37 °C in a humidified atmosphere with 5% CO_2_.

For CoQ determinations, cells were seeded in 12-well plates with an initial amount of 60,000 cells/well, and treated with 5 µM of kaempferol, ^13^C-kaempferol, ^13^C_6_*-[B-ring]-*kaempferol, or 1 µM 4HB for 48 h. In the previous publication where we described kaempferol as a novel CoQ precursor, experiments were made with 10 µM ^13^C-kaempferol [17]. However, the limited amount of ^13^C_6_*-[B-ring]-*kaempferol available led us to decrease the concentration used, although conditions are still in the range where kaempferol was reported to increase CoQ content [17]. Ethanol added to the control as vehicle was kept below 0.05% of the final volume. Cells were incubated under standard culture conditions (37 °C, 5% CO_2_). After the designated time, cells were washed twice with 1X phosphate-buffered saline (PBS), detached from the culture plates using trypsin-EDTA (Fisher Scientific, Waltham, MA, USA) and pelleted by low-speed centrifugation (approximately 1000× *g*). Supernatant was removed and cell pellets were stored at −20 °C until use.

### 4.3. Lipid Extraction

Cell pellets were resuspended in 100 µL of 1X PBS. Prior to lipid extraction, 10 µL aliquots was saved to quantify protein concentration using Bradford assay [32]. Then, dipropoxy-CoQ_10_ was added to the remaining 90 µL as internal standard. To start the extraction, two mL of methanol was added. The cell suspension was vortexed and two mL of petroleum ether was added. The upper petroleum ether layer (containing all non-saponifiable lipids, including CoQ) was transferred to a clean tube. Another two mL of petroleum ether as added to the original methanol layer, and samples were vortexed again. The top layer was removed, combined with the previous one, and the combined organic phase was dried under a stream of nitrogen gas. A series of CoQ_9_ and CoQ_10_ standards containing dipropoxy-CoQ_10_ were prepared and lipid extracted concurrently with the cell samples to construct CoQ_9_ and CoQ_10_ standard curves.

### 4.4. CoQ Analysis

Labeled and unlabeled CoQ_9_ and CoQ_10_ content from lipid extracts was analyzed using HPLC-MS/MS as described previously [17]. Briefly, samples were resuspended in 200 µL of ethanol containing 1 mg/mL benzoquinone in order to oxidize all the lipids prior to the analysis. A 4000 QTRAP linear MS/MS spectrometer from Applied Biosystems (Foster City, CA, USA) was used. Applied Biosystem software, Analyst version 1.4.2, was used for data acquisition and processing. Chromatographic separation was performed on a Luna 5 μm PFP(2) 100A column (100 × 4.6 mm, 5 μm; Phenomenex, Torrance, CA, USA) using a mobile phase composed of 90% solvent A (95:5 mixture of methanol:isopropanol containing 2.5 mM ammonium formate) and 10% solvent B (isopropanol containing 2.5 mM ammonium formate) at a constant flow rate of 1 mL/min. All samples were analyzed in multiple reaction monitoring mode. Transitions used were: *m/z* 795.6/197.08 (CoQ_9_+H), *m/z* 812.6/197.08 (CoQ_9_+NH_3_), *m/z* 801.6/203.08 (^13^C-CoQ_9_+H), *m/z* 818.6/203.08 (^13^C-CoQ_9_+NH_3_), *m/z* 863.6/197.08 (CoQ_10_+H), *m/z* 880.6/197.08 (CoQ_10_+NH_3_), *m/z* 869.6/203.08 (^13^C-CoQ_10_+H), *m/z* 886.6/203.08 (^13^C-CoQ_10_+NH_3_), *m/z* 919.7/253.1 (dipropoxy-CoQ_10_+H), *m/z* 936.7/253.1 (dipropoxy-CoQ_10_+NH_3_). The area of each peak, normalized with the correspondent standard curve and the internal standard, was referred to the initial amount of protein.

### 4.5. Statistical Analysis

Data shown in this work represent mean ± standard deviation (SD). Statistical analyses and graphics were performed with Graphpad Prism 8 (Graphpad Software Inc., San Diego, CA, USA). Differences in CoQ content in comparison with the control were analyzed using parametric one-way ANOVA, correcting for multiple comparisons with Dunnett’s post-test. Significant differences were referred as * *p* < 0.05, ** *p* < 0.01, *** *p* < 0.001, and **** *p* < 0.0001.

## 5. Conclusions

Our results show that kidney cells can cleave the B-ring of the dietary flavonol kaempferol to produce potential ring precursors of CoQ biosynthesis, most likely 4HB. This metabolism of kaempferol augments CoQ biosynthesis and increases CoQ content. This ability of kaempferol could potentially be used in the design of more efficient supplements to alleviate the symptoms of CoQ deficiencies in aging and disease. Additional physiological studies will be necessary to confirm the effectiveness of kaempferol supplementation to potentiate ubiquinone biosynthesis at the whole organism level.

## Figures and Tables

**Figure 1 molecules-25-02955-f001:**
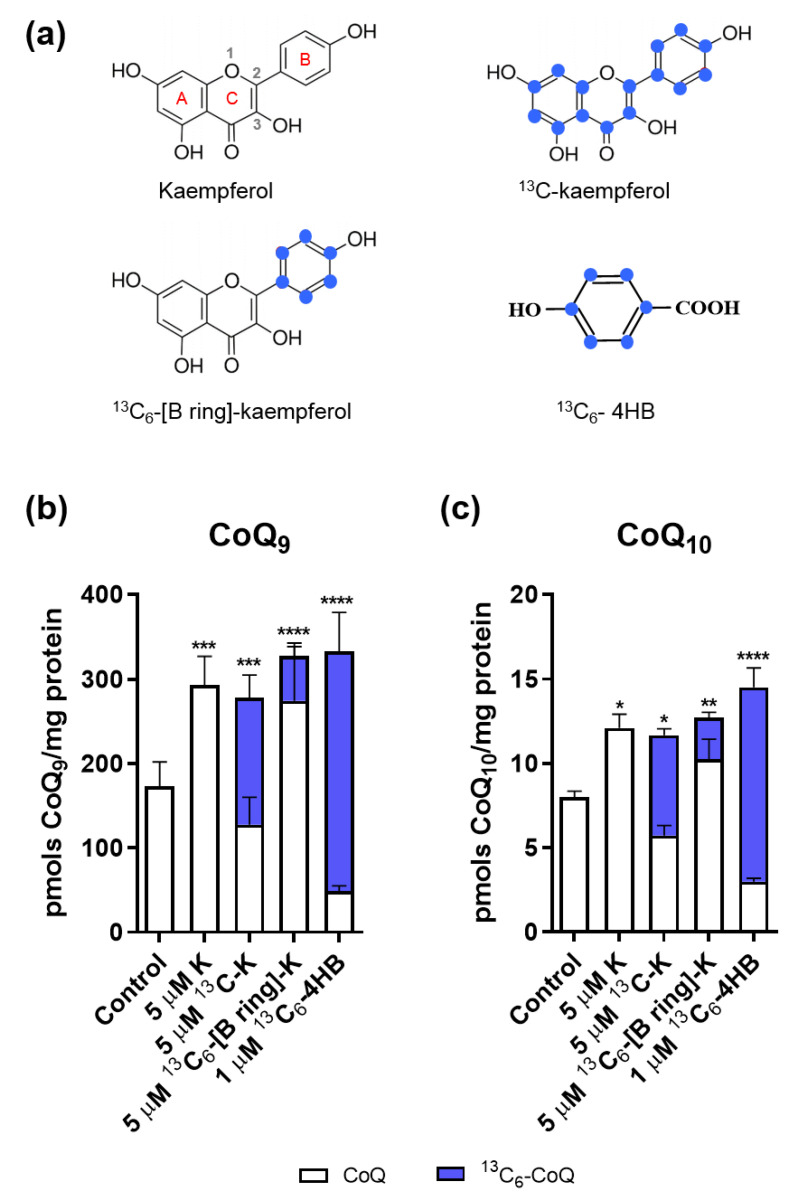
Kaempferol B-ring enters the coenzyme Q (CoQ) biosynthesis pathway. (**a**) Scheme representing the different ^13^C labels of kaempferol (K) and 4HB. Blue dots represent ^13^C-labeled carbons. Kaempferol A-B-C rings, as well as the most relevant positions in the molecule, are represented in the first kaempferol structure. (**b**) CoQ_9_ and ^13^C-CoQ_9_ content. (**c**) CoQ_10_ and ^13^C-CoQ_10_ content. In (**b**,**c**), mouse kidney proximal tubule epithelial (TKPTS) cells were treated with the different compounds for 48 h. Data represent mean ± SD of eight biological replicates coming from two independent experiments. Differences between total CoQ (CoQ+^13^C-CoQ) and the control are represented as * (*p* < 0.05), ** (*p* < 0.01), *** (*p* < 0.001) and **** (*p* < 0.0001).

**Figure 2 molecules-25-02955-f002:**
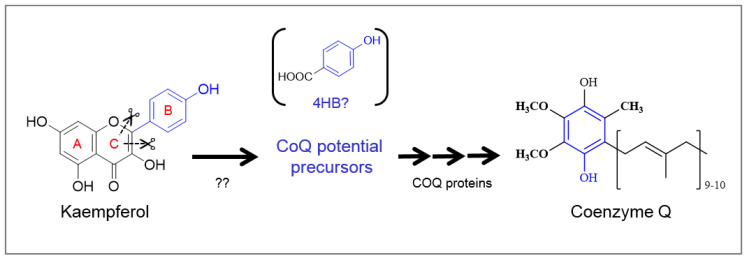
Model of kaempferol cleavage in kidney cells. Inside the cells, the B-ring of kaempferol is cleaved via a presumptive peroxidative mechanism, to produce CoQ precursors (most likely 4HB) that will enter the CoQ biosynthetic pathway to produce CoQ. The additional supply of this ring precursor is able to increase CoQ content in cells.

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
