# Peer review of "Metabolism of the Flavonol Kaempferol in Kidney Cells Liberates the B-ring to Enter Coenzyme Q Biosynthesis"

_molecules, 2020, doi:10.3390/molecules25132955_

Round 1

Reviewer 1 Report

The article by  Fernández-del-Río et al. presents some very interesting data relative to kempferol as a precursor in CoQ biosynthesis. The data confirm some observations of polyphenols use in the CoQ biosynthetic machinery but notably in this article the authors demonstrate that the same process applies to mammalian cells. The results are of major interest in light of the limited bioavailability of exogenous CoQ suggesting some promising alternative strategies for enhancing cellular CoQ content. 

Author Response

We thank the reviewer for the positive synopsis of our work. We have carefully checked the text and corrected all the misspellings.

Reviewer 2 Report

A vy interesting study showing that the flavonoid Kaempferol acts as a precursor of the benzoquinone ring of Coenzyme Q in animal cells. The study has been conducted onkidney cells in vitro and showed that the B ring of the flavonoid is totally incorporated into CoQ, probably acting as a source  of 4.hydroxybenzoate. Th study is important because it may be a premise to enhance the biosynthesis of CoQ, whose exogenous uptake is limited.

Author Response

We thank the reviewer for this favorable summary of our research.

Reviewer 3 Report

Molecules (Manuscript ID: molecules-827988), Comments to the Authors:

Title: Metabolism of the flavonol kaempferol in kidney cells liberates the B-ring to enter coenzyme Q biosynthesis

Comments

The submitted manuscript discussed the labeling of kaempferol specifically in the B-ring was isolated from Arabidopsis plants. Kidney cells treated with this compound incorporated the B-ring of kaempferol into newly synthesized CoQ, suggesting that the B-ring is metabolized via a mechanism described in plant cells.

Despite the presented results, I think the manuscript cannot be accepted for publication. The results are preliminary, and the concept was proven in previous reports. Also, kaempferol is poorly absorbed and most of the in vitro results do not reflect any true situation. The authors need to prove their concept in vivo.

Author Response

We agree with the reviewer that in vivo experiments are absolutely necessary to prove the potential use of kaempferol in supplements to treat CoQ deficiencies. We have included a sentence in Section 5, Conclusions stating (lines 254-256), “Additional physiological studies will be necessary to confirm the effectiveness of kaempferol supplementation to potentiate ubiquinone biosynthesis at the whole organism level.” However, such experiments are beyond the scope of the present Communication. Per the Instructions for authors of Molecules, the goal of Communications is to present “preliminary, but significant, results” that, in our case, help to better understand the relationship between kaempferol and coenzyme Q. Our results add a new layer of information to previous publications by showing that the B ring of kaempferol is the part of the molecule that enters coenzyme Q biosynthesis and that this metabolism resembles what was observed in plants, suggesting a conserved mechanism in mammalian cells.

Round 2

Reviewer 3 Report

Molecules (Manuscript ID: molecules-827988), Comments to the Authors:

Title: Metabolism of the flavonol kaempferol in kidney cells liberates the B-ring to enter coenzyme Q biosynthesis

Comments

After reading the authors response to my comments, I still believe that the presented results do not merit publication. As I mentioned before, the results are preliminary, do not answer important question in the field and the concept was already proven in previous publications. I believe the submitted paper should be rejected.